# The impact of extreme heat on lake warming in China

Weijia Wang[1,2,3], Kun Shi [1,2] ✉, Xiwen Wang[1,4], Yunlin Zhang[1,2,3], Boqiang Qin [1,4], Yibo Zhang[1] & R. Iestyn Woolway[5]

Global lake ecosystems are subjected to an increased occurrence of heat extremes, yet their impact on lake warming remains poorly understood. In this study, we employed a hybrid physically-based/statistical model to assess the contribution of heat extremes to variations in surface water temperature of 2260 lakes in China from 1985 to 2022. Our study indicates that heat extremes are increasing at a rate of about 2.08 days/decade and an intensity of about 0.03 °C/ day·decade in China. The warming rate of lake surface water temperature decreases from 0.16 °C/decade to 0.13 °C/decade after removing heat extremes. Heat extremes exert a considerable influence on long-term lake surface temperature changes, contributing 36.5% of the warming trends within the studied lakes. Given the important influence of heat extremes on the mean warming of lake surface waters, it is imperative that they are adequately accounted for in climate impact studies.

There is overwhelming evidence that heat extremes (e.g., heatwaves), one of the perilous consequences of climate change, are exhibiting an alarming increase in frequency and intensity worldwide[1]. These extreme events pose severe threats to natural ecosystems, socio-economic stability, and human well-being, leading to irreversible ecological and societal impacts[2]. Their ramifications encompass not only direct human and biological mortality but also exacerbation of other disasters such as wildfires, mental health issues, and agricultural losses[2,3]. Numerous instances highlight the devastating consequences of heat extremes. For example, the 2003 heatwave in Europe, considered the hottest summer in the past five centuries[4], claimed the lives of at least 70,000 people[5]. In Western Russia, heat extremes in 2010 resulted in 500 fires around Moscow and a substantial 30% decline in grain harvest[6]. More recently, during late-June 2021, the Pacific Northwest region of Canada and the United States experienced extreme heat, leading to hundreds of deaths[7]. Additionally, the summer of 2022 witnessed unprecedented heat extremes sweeping through various parts of the world, including London and Shanghai[8], triggering widespread fires in France, Spain, Greece and Germany. In China alone, over 400 cities endured extreme heat[9].

Lakes, as crucial and vulnerable components of the Earth's eco-system, bear significant impacts during heat extremes[10]. The substantial increase in lake surface water temperature (LSWT) induced by these events can rapidly disrupt the physical, chemical, and biological properties of a lake, thereby perturbing the entire lake ecosystem with potentially irreversible consequences[11]. While the long-term warming of LSWT, and the escalating occurrence of lake heatwaves, as well as their knock-on effects[11–13] have been investigated extensively and documented globally[12,13], a critical knowledge gap persists regarding the specific effects of extreme heat events on lake warming and the increased occurrence of lake heatwaves.

Critical freshwater resources such as lakes are distributed across diverse climatic and geographic environments (Supplementary Fig. 1), potentially influencing their responses to heat extremes[14]. In China, lakes larger than 1 km² cover a substantial area of approximately 93,723 km² and play an irreplaceable role in flood and drought prevention, water purification, and biodiversity conservation. Moreover, they serve as a vital source of municipal drinking water, supplying 51.0% of the population in the eastern region, and holding significant cultural and economic importance[15]. However, despite their ecological

[1]Taihu Laboratory for Lake Ecosystem Research, State Key Laboratory of Lake Science and Environment, Nanjing Institute of Geography and Limnology, Chinese Academy of Sciences, Nanjing 210008, China. [2]University of Chinese Academy of Sciences, 100049 Beijing, China. [3]College of Nanjing, University of Chinese Academy of Sciences, Nanjing 211135, China. [4]School of Geography & Ocean Science, Nanjing University, Nanjing, China. [5]School of Ocean Sciences, Bangor University, Menai Bridge, Anglesey, UK. ✉e-mail: kshi@niglas.ac.cn

and socio-economic significances, the response of these lakes to climate change, particularly in relation to heat extremes, has received comparatively little attention[16]. Consequently, there exists a compelling need to enhance our understanding of the key processes and mechanisms that underlie the effects of heat extremes on lakes in China, a region characterized by highly variable spatial lake responses. To address this knowledge gap, we employed long-term daily simulations to reveal the spatial and temporal variation patterns of LSWT across China and quantified the contribution of heat extremes to lake warming.

## Results

The frequency of air temperature extreme heat events in China has followed an increasing trend, albeit with notable regional variability. Over the course of our 38-year investigation from 1985 to 2022, China experienced an average of approximately 454 days of extreme heat (see "Methods"), with marked spatial variability (Fig. 1; Supplementary Table 1). The southern regions of China exhibited the highest frequency of heat extremes, with a total of 552 days, while the eastern coastal and western regions of China had the lowest frequency, with a

minimum of 414 days. The national trend of extreme high temperatures displayed an overall increase, with a rate of approximately 2.08 days/decade, and the most substantial increase occurred in the southwest at a rate of 6.09 days/decade. The spatial pattern of total cumulative heat (see Methods) exhibited distinct characteristics. The Inner Mongolia-Xin Jiang Lake Region and North-east Plain Lake Region showed the highest cumulative heat level, reaching up to 785 °C, while the southern region of China exhibited the lowest level, close to 175 °C. The annual cumulative heat showed an increasing trend across the country, particularly notable in the Inner Mongolia-Xin Jiang Lake Region, with a rate of 6.93 °C/decade. The intensity of extreme heat followed a similar spatial pattern to the accumulated heat, showing an overall increasing trend of approximately 0.03 °C/day·decade. The highest trend of heat intensity was observed in the Mongolia-Xin Jiang Lake Region, at approximately 0.29 °C/day·decade, while the Tibetan Plateau Lake Region and the North-east Plain Lake Region had the lowest trend, at −0.11 °C/day·decade.

Across the studied regions, lakes were exposed to extreme high surface air temperatures (SAT) for an average of 459 ± 19 days (min: 409 days, max: 518 days) between 1985 and 2022 (Supplementary Fig. 2).

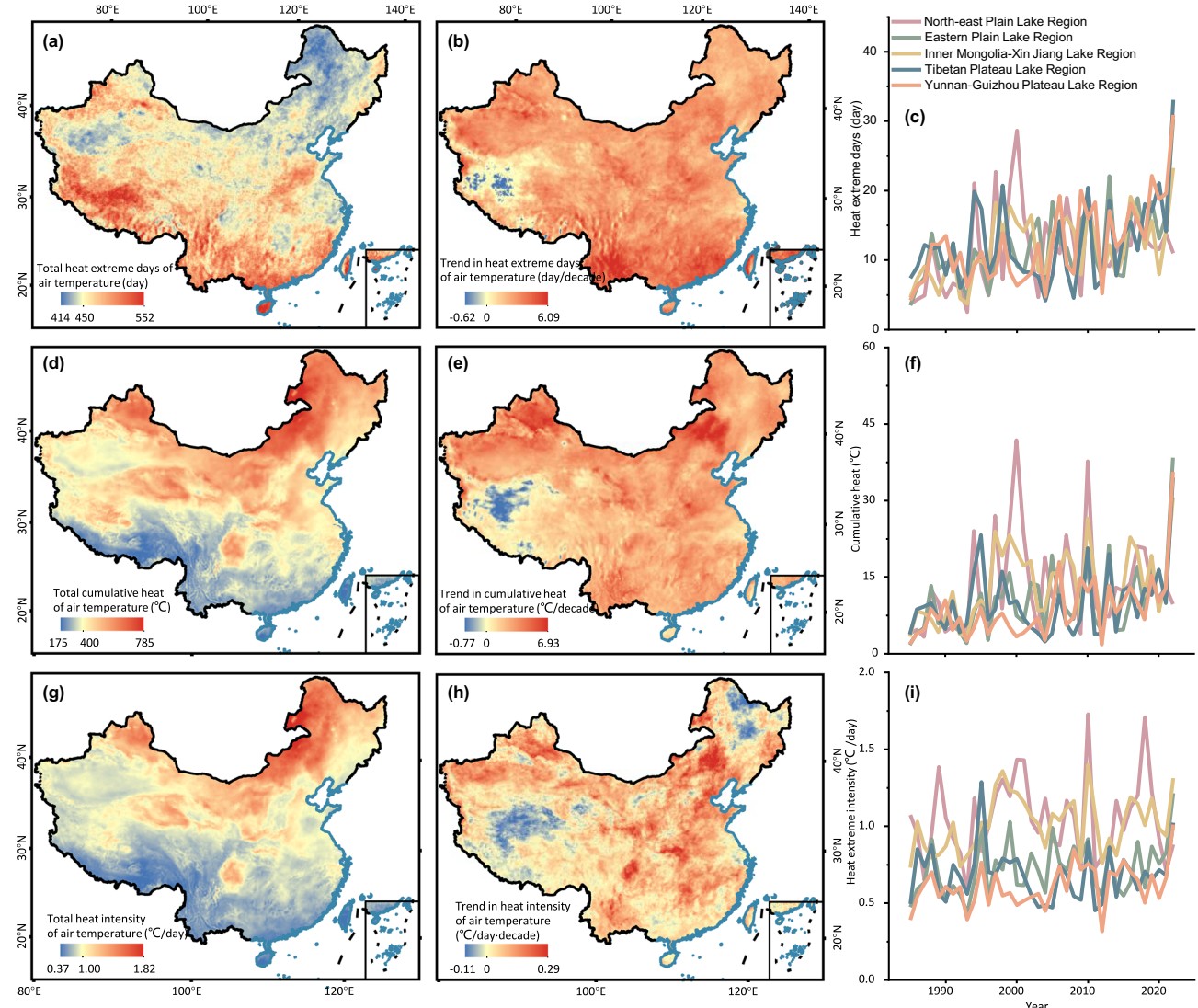

**Fig. 1 | Spatial and temporal variations in heat extremes of air temperature. a–c** Show the total number of heat extreme days from 1985 to 2022, the decadal trend, and the annual variation in the five lake regions (Eastern Plain Lake Region, North-east Plain Lake Region, Inner Mongolia-Xin Jiang Lake Region, Tibetan Plateau Lake Region, and Yunnan-Guizhou Plateau Lake Region), respectively. **d–f** Indicate the cumulative heat for 38 years, the decadal trend, and the annual variation in the five lake regions, respectively. **g–i** Represent the corresponding information of heat intensity. Source data are provided as a Source Data file.

The frequency of heat extremes in SAT exhibited an overall increasing trend, with a rate of 1.89 days/decade. The Yunnan-Guizhou Plateau Lake Region had the highest rate at 5.45 days/decade, while the Tibetan Plateau Lake Region displayed the lowest rate at −1.25 days/decade.

Extreme heat events have made a substantial contribution to the long-term warming trend observed in China. Analysis of the data from European Center for Medium-Range Weather Forecasts (ECMWF) Reanalysis v5 - Land (ERA5-Land) revealed a warming trend in air temperatures from 1985 to 2022, with an average increase of 0.32 ± 0.10 °C/decade (Supplementary Fig. 3). Notably, the Inner Mongolia-Xin Jiang Lake Region exhibited a particularly rapid warming trend of 0.68 °C/decade, while the Tibetan Plateau Lake Region showed a negligible variation and even a cooling trend of −0.07 °C/decade. Examining the average trend of the SAT during the same period, we found it to be 0.29 ± 0.11 °C/decade (min: −0.07 °C/decade, max: 0.68 °C/decade). However, upon excluding the heat extremes (see Methods), the overall air temperature trend across China decreased slightly to 0.17 ± 0.08 °C/decade (min: −0.10 °C/decade, max: 0.43 °C/decade). The areas most affected by heat extremes in China, as indicated by the difference between trends of air temperature and air temperature after the removal of heat extremes (RHAT), were the Inner Mongolia-Xin Jiang Lake Region and the Eastern Plain Lake Region, with a difference of 0.22 °C/decade. Similarly, after excluding heat extremes, the average trend of SAT changed from 0.29 ± 0.11 °C/decade (min: −0.04 °C/decade, max: 0.51 °C/decade) to 0.22 ± 0.19 °C/decade (min: −0.04 °C/decade, max: 0.51 °C/decade).

These heat extremes exerted a strong impact on lake temperatures. Our long-term daily simulations of LSWT revealed significant differences in the occurrence of extremes before and after removing heat extremes from SAT (RHSAT) in the forcing field of Air2Water. Upon excluding heat extremes from SAT, the average number of days with extreme heat in LSWT decreased substantially from 382 to approximately 314 days. Furthermore, the mean cumulative heat decreased from 172.98 °C to 66.65 °C, and the heat intensity decreased from 0.43 °C/day to 0.19 °C/day (Supplementary Fig. 4). When the annual cumulative heat of SAT increased by 1 °C, the intensity of the heat extremes of LSWT increased by approximately 0.01 °C/day, the cumulative heat increased by about 0.45 °C, and the number of days of heat extremes increased by about 0.66 days. With 1 day increase in heat extreme days of SAT, the heat extreme days in LSWT increased by about 0.93 days and the cumulative heat gained about 0.51 °C. An increase of 1 °C/day in the intensity of heat extremes in SAT would result in an increase in that of LSWT of about 0.42 °C/day (Supplementary Fig. 5). Additionally, the national average summer LSWT during the same period decreased by 0.22 °C after removing heat extremes from SAT in the forcing field of Air2Water (Supplementary Fig. 6). The analysis indicated an overall increasing trend with an average of 0.16 °C/decade. Lakes in the Eastern Plain Lake Region experienced the greatest warming, with a rate of 0.22 °C/decade, while lakes in the Tibetan Plateau Lake Region showed the least warming, with a rate of 0.08 °C/decade (Fig. 2). After removing heat extremes, the average warming rate of LSWT (RHLSWT) decreased to 0.13 °C/decade. Our analysis suggests that the documented extremes in SAT resulted in an increase in national LSWT of about 0.03 °C/decade. The difference between LSWT and RHLSWT exhibited an increasing trend, growing from 0.10 °C in 1985 to 0.47 °C in 2022. This indicated that changes in heat extremes, in addition to the context of global climate change, contribute to the increase in LSWT, as heat extremes were removed without altering the global warming trend. The trend in SAT due to heat extremes (trend of SAT minus trend of RHSAT) across lakes in China was approximately 0.07 °C/decade, which was slightly higher than that of LSWT by 0.03 °C/decade. The difference in annual mean values of SAT attributable to heat extremes increased at a rate of 0.04 °C/decade above that of LSWT. Moreover, the increase in the trend of days with heat extremes in SAT was 1.75 days/decade, which

was about 0.23 days/decade higher than that in LSWT. Although the total number of heat extreme days in SAT accounted for approximately 3% of the total study period (about 459 days), the contribution of heat extremes to the LSWT in China from 1985 to 2022 (see Methods section for calculation) reaches up to 36.5%. This highlights the unexpectedly large impacts of heat extremes on LSWT, despite their low frequencies and short durations. It further indicated that short-term occurrences of heat extremes can profoundly influence lakes on seasonal, annual, and even longer time scales.

In the summer of 2022, China suffered the most intense and prolonged extreme heat event since 1961[17]. The average number of extreme heat days in China in 2022 was about twice the level of 1985-2021, and the average cumulative heat was approximately three times higher than before. The average heat intensity in 2022 was about 25% greater than that in the past 37 years. Considering the severity of heat extremes in 2022, we repeated the numerical experiment for 1985-2021, and the results showed that trends of heat extreme days, cumulative heat, and heat intensity of SAT in China for 1985-2021 were 2.05 days/decade, 1.88 °C/decade and 0.04 °C/day·decade, respectively. For 1985–2022, they were 2.08 days/decade, 1.95 °C/decade, and 0.04 °C/day·decade, respectively. The average trend of LSWT was 0.16 °C/decade for 1985-2021 and 0.15 °C/decade for 1985–2022. Trends in heat extreme metrics and LSWT did not differ much between the cases including and excluding 2022; this is because we used the Theil-Sen method for the calculation of long-term trends, which is robust to outliers such as 2022. The average contribution of heat extremes to LSWT was 40.4% for 1985-2021 and 36.5% for 1985–2022. Therefore, the severe heat extreme in 2022 did not bias our conclusions.

## Discussion

Evaporation and precipitation emerged as primary drivers influencing the temporal change in the difference between LSWT and RHLSWT (Supplementary Fig. 7). Specifically, higher rates of evaporation and greater precipitation were associated with a more pronounced impact of heat extremes on LSWT[10]. Regarding spatial factors, two key variables were identified as having a predominant influence on the contribution of heat extremes to LSWT: shortwave radiation and Secchi disk depth (SDD). The former is often considered one of the primary external factors influencing lake surface temperatures[18]. Moreover, as SDD decreases, the amount of solar shortwave radiation energy penetrating the lake's surface layer increases, often leading to a rapid rise in LSWT[12].

Rapid increases in LSWT due to heat extremes can have catastrophic consequences for aquatic lives, leading to mortality and the potential for algal blooms that can shift a lake from clear to turbid[19,20]. Even subtle changes in the physical or chemical processes within a lake due to sudden LSWT fluctuations can have substantial ecological impacts, threatening the survival of aquatic lives, especially when water levels and oxygen concentrations decreased[19]. One of the most severe heat extremes in the last 60 years swept through China during the summer of 2022, resulting in a remarkable increase in LSWT of 1.63 °C compared to the period from 2000 to 2021[17]. This extreme event triggered a series of catastrophes, including a decrease in water level, a dramatic reduction in water surface area, a massive die-off of aquatic organisms, and a shortage of water and electricity supply for the residents of surrounding cities[17].

The investigation into the magnitude and underlying processes of the impact of heat extremes on LSWT will significantly enhance our understanding of lake ecosystem responses to climate change and lay a theoretical foundation for future lake system management. However, current climate models face challenges in accurately projecting future extreme events and may underestimate future changes in heat extremes[21]. Thus, future projections of LSWT could be underestimated by at least 36.5%. Better representation of heat extremes by climate models is critical for improving our management of heat extremes on

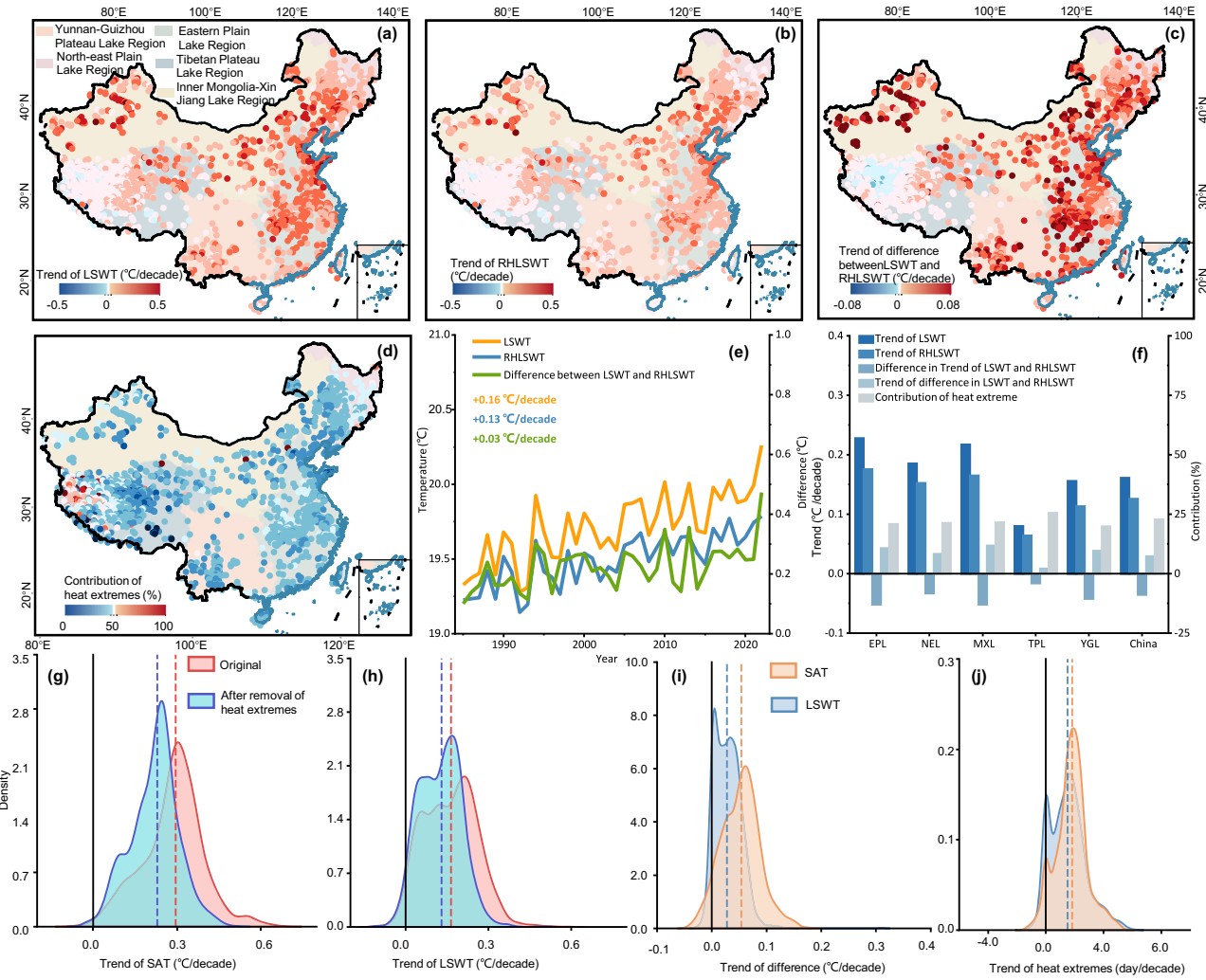

**Fig. 2 | Contribution of heat extremes to lake surface water temperature (LSWT). a** Trend of LSWT. **b** Trend of LSWT simulated by removing heat extremes from lake surface air temperature (SAT) in Air2Water (RHLSWT). **c** Trend of differences between LSWT and RHLSWT. **d** Contribution of heat extremes to lake warming. **e** Average LSWT, RHLSWT and differences between LSWT and RHLSWT from 1985 to 2022. **f** Histogram of trend and contribution, where difference in trend of LSWT and RHLSWT represents the trend of RHLSWT minus the trend of LSWT.
**g** Probability density of trend in SAT (red) and SAT after removal of heat extremes (RHSAT) (cyan). **h** Probability density of trend in LSWT (red) and RHLSWT (cyan). **i** Probability density of trend in difference between LSWT and RHLSWT (blue) and difference between SAT and RHSAT (orange). **j** Probability density of trend in heat extremes of LSWT (blue) and SAT (orange). The dashed line indicates the median value of the dataset with the corresponding color. Source data are provided as a Source Data file.

ecosystem services. Considering the sensitivity of lake ecosystems to heat extremes, the observed expansion of such events is expected to exert substantial impacts on aquatic ecosystems[22]. In response to this emerging reality, it is imperative for management departments to establish robust systems for monitoring and predicting heat extremes. Additionally, measures should be taken to enhance the resilience of lake ecosystems to climate risks and implement effective disaster mitigation and prevention strategies. This proactive approach is essential for safeguarding the integrity and functionality of lake ecosystems in the face of increasing heat extremes and their potential ecological disruptions.

## Methods

### Hybrid model of lake water surface temperature
The Air2Water model is a hybrid physically-based/statistical model that mathematically simplifies all heat flux components at the lake-atmosphere interface, including the shortwave radiation, longwave radiation, and diffusion terms, to obtain a simple ordinary differential equation, allowing LSWT to be appropriately modeled using only surface air temperature observations as a reliable substitute for overall

external forcings[23]. The Air2Water model has been widely used in regional and even global studies[24] due to its advantages of easy access to data relative to other hydrologic models, low reference data requirements relative to machine learning models, and the combination of physical processes and statistical methods[25]. The daily 2 m air temperature at a grid resolution of 0.1° from ERA5-Land was used to run Air2Water from 1985 to 2022. By utilizing air temperature as input and satellite-derived or in situ LSWT as a reference, the eight model parameters of Air2Water were calibrated by optimizing a metric of model performance using an automated optimization process (particle swarm optimization)[26].

Given the challenges of acquiring complete in situ LSWT data over a long time period, we chose the Landsat land surface temperature product which has a data record of more than 38 years to be the satellite-derived LSWT. Evidence has shown that the product has satisfactory accuracy at the water surface with an average bias of −0.3 °C and an RMSE (Root Mean Square Error) of 1.1 °C when compared with in situ measurements[24]. It has been widely used in various academic disciplines due to its advantages of easy access, possession of long-time records, and high accuracy[27–29].

We conducted an experiment to demonstrate the necessity of uniformly distributed satellite data with complete coverage of the entire simulation period as a reference in Air2Water (Supplementary Fig. 8). In this experiment, a complete sequence of satellite data from 2013-2019 and satellite data from 2013-2017 were used as references, respectively, in combination with the daily air temperature to calculate the daily LSWT from 2013 to 2019. The results showed that despite the comparisons of the total data were comparable, the simulated LSWT without satellite data ($R^2 = 0.92$ and MAE (Mean Absolute Error) = 1.83 °C) were subject to greater errors than those with satellite data as a reference ($R^2 = 0.94$ and MAE = 1.59 °C) in the validation of 2018-2019. Moreover, the inclusion of satellite data with relatively homogeneous time intervals in the whole time series can make the overall accuracy more stable[24].

We collected tens of thousands of in situ LSWT observations from six lakes (Lake Erhai, Lake Hulunhu, Lake Namco, Lake Luguhu, Lake Qiandaohu and Lake Taihu) (Supplementary Table 2) of varying sizes, elevations, latitudes, and depths to verify the confidence of our simulations. These data were recorded at daily to monthly intervals spanning the years 1993 to 2018. Satisfactory accuracy ($R^2 \geq 0.93$ and MAE <2.00 °C) was obtained in the comparison of simulated results from Air2Water with the in situ data (Supplementary Fig. 9). The best performer among them is Lake Taihu, which was in a position to obtain an excellent accuracy of $R^2 = 0.98$ as well as MAE = 1.88 °C with the validation of 9289 pairs of data. The least accurate among these is Lake Hulunhu, which also achieved outstanding results with $R^2 = 0.93$ and MAE = 1.99 °C.

We have compared the Air2Water model to another popular lake model, FLake (Freshwater Lake model) and one of the most widely used machine learning models (artificial neural network, ANN) in terms of its effectiveness in simulating LSWTs. The FLake model is a one-dimensional bulk model based on the concept of self-similarity, where the vertical profiles of the mixed and thermocline layers are described by the respective shape functions, resulting in a low cost of computation[30]. In addition, it contains few lake-specific parameters for the model and does not require extensive calibration[13]. The meteorological forcing data used in FLake were obtained from ERA5-Land, with initial parameter settings referenced in ref. 1, the depths of the individual lakes used were the average depths from the HydroLAKES database[31], and the lake ice albedo was set to 0.6. A variety of machine learning models for simulating LSWT have been compared in 2022 and the results showed that ANN is the most used and successful machine learning algorithm for LSWT prediction[32]. The tansig function and purelin function were set as the hidden layer transfer function and output layer function in ANN, respectively. The learning rate, target error and momentum were set to 0.001, 0.0001 and 0.95, respectively after making adjustments through a step-by-step grid strategy. The optimal number of hidden layers was determined to be 3 after preliminary experiments. We modeled each lake individually using SAT, year, and DOY (day of the year) as input parameters and LSWT as output to the ANN. We compared the simulation results of FLake and ANN for the same air temperature and observed LSWT from Landsat as a reference in six lakes for which in situ data were available. The results showed that the average $R^2$ and MAE between simulations and in situ LSWT for FLake were 0.74 and 3.62 °C, respectively (Supplementary Fig. 10). For ANN, the values of the two metrics were 0.87 and 2.59 °C, respectively. Both FLake and ANN simulations did not show comparable performance with Air2Water ($R^2$ and MAE were 0.96 and 1.38 °C, respectively), demonstrating the applicability of Air2Water model for the present research.

### Identifying heat extremes and modeling LSWT before and after the removal of heat extremes
The probability of the presence of water at a spatial resolution of 30 m between 1985 and 2020 in the global surface water occurrence (GSWO

v1.3) dataset was used to generate a mask of water surfaces. The fraction of GSWO ≥ 95% and the polygons of lakes ≥1 km² in Hydro-LAKES were overlaid, and the bays misidentified as lakes were removed to finally determine the extent of studied lakes, with a total number of 2260 (Source Data).

We selected the long-term continuous high spatial resolution (60–120 m) surface temperature data from Landsat 5, 7, and 8 provided by the National Aeronautics and Space Administration (NASA) and the U.S. Geological Survey (USGS) as the LSWT. Variations exist between sensors for Landsat 5, 7, and 8, therefore we selected thermal infrared (TIR) data for Landsat 5 from 1985 to 2011, Landsat 7 from 1999 to 2018, and Landsat 8 from 2013 to 2022 to constitute the three LSWT datasets. A lake-specific LSWT was calculated as the average temperature of all pixels within the lake that had been resampled to a resolution of 60 m. To eliminate the outliers, we divided each dataset into 12 sub-datasets from January to December, excluded values with a distance of more than three standard deviations from the median in each sub-dataset, and repeated the operation twice. The datasets covering 2013–2022 were used as a basis, and linear models were constructed using the seasonal average data pairs within the overlapping time to correct the other two datasets to finally obtain the complete LSWT dataset from 1985 to 2022.

Each day in the extended summer season (June-September) that exceeds the 90th percentile climatology of the corresponding calendar day is defined as a heat extreme day[33]. The 90th percentile climatology was produced by computing the daily 90th percentile of air temperature or LSWT using an 11-day window centered on the day of the year over a partially moving baseline. The partially moving baseline is fixed for the first and last 31 years of 1985–2022 and moves in the middle. Specifically, the heat extremes for 1985-2000 were calculated using a fixed baseline from 1985 to 2015, for 2007-2022 a fixed baseline from 1992 to 2022, while for 2001–2006, a 31-year moving baseline centered on the year in question was used[33,34]. In this study, cumulative heat was defined as the accumulation of air temperature or LSWT above 90th percentile climatology for each heat extreme day during the season of interest. Heat intensity was defined as the average temperature anomaly per day of extreme heat during the season of interest, i.e., the cumulative heat divided by the extreme heat days. The RHSAT was obtained by replacing heat extreme days with the climatological mean calculated from the partially moving baseline. The RHLSWT was simulated by Air2Water using RHSAT with the optimal parameters previously calculated in Air2Water using SAT as the forcing field.

### Contribution of heat extremes
Linear fit functions for the summer average values of each lake were constructed separately for the original LSWT and RHLSWT using the Theil-Sen method for 1985–2022 (Supplementary Fig. 11). The difference between the values of the two linear functions in 2022 was recorded as the difference in 2022 (labeled as BC in the figure). An initial temperature was set for each lake, expressed as the value of the linear function of RHLSWT in 1985 for LSWT with a positive trend, and as the value of the linear function of LSWT in 1985 for LSWT with a negative trend. The total change was defined as the difference between the value of the linear function in 2022 and the initial value and was plotted as AC for a positive trend in LSWT and BC for a negative trend. The contribution of heat extremes to LSWT was defined as the difference in 2022 divided by the total change, that is, expressed as BC/AC when LSWT has a positive trend and as BC/AB when LSWT is negative.

### Importance assessment
In this study, the feature importance was assessed using a random forest algorithm. Spatially, the importance of 17 factors such as lake elements (lake area, depth, volume, SDD), geographical conditions

(latitude, elevation), climate (precipitation, evaporation, longwave radiance, shortwave radiance, U component of wind speed, V component of wind speed, and humidity), and human activities (population, impervious surface, normalized difference vegetation index (NDVI), and gross domestic product (GDP)) were assessed for the contribution of heat extremes in 2260 lakes. Temporally, the impact of the above seven meteorological elements on the intensity of extreme heat (difference between the mean values of summer LSWT and RHLSWT) was assessed on an annual scale.

## Data availability

The ERA5-Land data used in this study are available at https://cds.climate.copernicus.eu/cdsapp#!/dataset/10.24381/cds.e2161bac?tab=overview; Landsat LSWT data are available at https://developers.google.com/earth-engine/datasets/catalog/LANDSAT_LT05_C02_T1_L2, https://developers.google.com/earth-engine/datasets/catalog/LANDSAT_LE07_C02_T1_L2, https://developers.google.com/earth-engine/datasets/catalog/LANDSAT_LC08_C02_T1_L2; HydroLAKES dataset is available at https://www.hydrosheds.org/pages/hydrolakes; GSWO are available at https://developers.google.com/earth-engine/datasets/catalog/JRC_GSW1_3_GlobalSurfaceWater. The data of LSWT and RHLSWT generated in this study are available at https://doi.org/10.11888/Terre.tpdc.300801, Source data are provided with this paper.

## Code availability

Air2Water source codes are available at https://github.com/marcotoffolon/air2water, FLake source codes are available at http://www.flake.igb-berlin.de/. The source codes used in this study are publicly available at https://doi.org/10.5281/zenodo.10214063[35].

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

## Acknowledgements

This study was supported by the National Natural Science Foundation of China (U22A20561 and 41922005) to K.S., the Tibetan Plateau Scientific Expedition and Research Program (2019QZKK0202) to K.S., the National Key Research and Development Program of China (2022YFC3204100) to Y.Z., the NIGLAS foundation (E1SL002) to K.S., and the UKRI Natural Environment Research Council (NERC) Independent Research Fellowship (NE/T011246/1) to R.I.W.

## Author contributions

W.W. conducted data analysis, designed visualizations, and wrote the original manuscript. K.S. proposed the concepts, supervised the project, and helped edit the manuscript. X.W. helped with data processing, validation, and manuscript editing. Y.Z. contributed to research design and editing. B.Q. refined the concepts and helped with editing. Y.Z. contributed to data processing and validation. R.I.W. supervised the project and helped edit the manuscript.

## Competing interests

The authors declare no competing interests.
