## [Peer Review File · Nature Communications]

The impact of extreme heat on lake warming in ChinaREVIEWER COMMENTS

Reviewer #1 (Remarks to the Author):

Heat extremes have been widely awarded by the public, however, its impact on aquatic ecosystems is under studied. In this work, the authors present a quantitative analysis about heat wave impacts on long-term lake warming. While the finding that heat waves contribute to long-term lake warming in China is certainly important for the limnology community, there are several critical aspects of the analysis remains to be justified and improved. Therefore, some more analysis and major revision are necessary.

1. I notice that the title “Extreme heat drives unprecedented warming of lakes in China” is very similar to the title of the author’s last paper “A record-breaking extreme heat event caused unprecedented warming of lakes in China” which has been cited by this manuscript. This should be absolutely avoided in scientific publication. I think the first thing is to justify the unique value of this work in comparison to the author’s last paper, and why do not put the two works into single manuscript? The title should be modified accordingly.
2. Line 71-84: please specify in front of this paragraph that you are talking about the lake heat waves not heat waves of the atmosphere.
3. Line 85-86: I do not see the connections between this sentence and the context. Is there any relationship between heat waves in 2022 and the trend of heat wave over the last 38 years? Do you want to emphasize heat wave in 2022 is the only heat wave that exhibited inconsistent spatial pattern with the long-term trend of heat waves? Nevertheless, if you think it is necessary to emphasize the heat wave in 2022, please provide a figure or cite your heat wave paper about its spatial pattern, otherwise it is difficult for readers to follow your argument.
4. Line 104: “)” is missing.
5. Line 119-120: Do you mean “after removing heat extremes from SAT in your forcing field of Air2Water model”?
6. Line 125: I was confused by RHLSWT. Does it mean removing the heat waves from LSWT or removing heat waves from the SAT in your forcing field of Air2Water model? Removing heat waves from the SAT in the forcing field of Air2Water model is an effective way to count contribution of heat wave to long-term trend of lake warming.
7. Methods: How is the heat wave calculated? Did the authors use a temporarily constant threshold or a running-smoothed threshold for the heat waves? This work is to count the contribution of heat waves to long-term warming, and threshold of heatwaves should increase along with the increase of background temperature (i.e., mean temperature, because temperature generally follows a normal distribution). Therefore, using a running smoothed threshold to avoid mixture of long-term warming trend and heat wave in the calculation is necessary for this work.
8. Methods: Line 290-292, as follows the comments above, I would suggest that obtaining RHSAT by replacing heat extreme days with running smoothed temperatures instead of climatological mean of calendar days.
9. I see the linear trend analysis of heat waves count in 2022, however, heat waves in 2022 are two times stronger than heat waves in 1985-2021 based on Figure 1 a3 and b3. It is hard to say such strong heat

wave was totally attributed to long-term trend, instead, inter-annual variability plays a big role in the heat waves of 2022. If excluding 2022 from the heat wave analysis, the result could be quite different. It is suggested repeat the whole analysis without 2022 and check whether conclusion changed. I believe this will solidify the conclusion.

10. I would suggest the authors analyze the sensitivity of lake heat wave to the heat wave in the atmosphere, e.g., what is the changing magnitude of lake heat wave intensity corresponding one unit of the accumulative heat of atmospheric heat waves.

11. I see the authors use Landsat based lake surface temperature data product. Why not use MODIS based lake surface temperature product which has much higher temporal coverage (daily vs monthly from Landsat data) and accuracy, as the authors mentioned the satellite-based observation is very important for the accuracy of data driven model? If the authors worried MODIS data only available in 2001 to present and for large lakes, they can merge the two sets together. The authors may have some important arguments that Landsat-based only products are accurate enough for the analysis, and please justify it in the manuscript.

12. Reference 7: Instead of citing the research Briefing, I prefer to cite the original research paper, which is entitled "Disruption of ecological networks in lakes by climate change and nutrient fluctuations".

13. Supplementary Figure 1: I do not see the necessity of illustrating three climatic zones in this figure. There is very little text citing the three climatic zones. Instead, illustrating the extent of the five lake regions is necessary.

Reviewer #2 (Remarks to the Author):

Extreme heat drives unprecedented warming of lakes in China

This study used a data-driven model to assess the contribution of heat extremes to variations in surface water temperature of 2260 lakes in China from 1985 to 2022. The study is of great interests to the broad scientific community. I don't have any other comments except for two concerns about the methods, see below for the details.

Methods:

- (1) "The Air2Water model is a zero-dimensional thermal budget model". I doubt about this as the Air2Water model is not a 0D model. The model is used for LSWT. However, the inclusion of a time-varying δ allows this model to be classified as 0.5D, as it simulates the temporal evolution of the epilimnion. We described this model in detail in a recent review, which is now under review at *Reviews of Geophysics*. I suggest the authors rephrasing this sentence.
- (2) The reason for choosing the air2water model needs justification. I know that the air2water model is simple to use, however, there are some models that have proven to significantly

outperform this model, e.g., the stacked machine learning model in “A stacked machine learning model for multi-step ahead prediction of lake surface water temperature”. I concern about this point also because of the relatively large MAE values in this study and I doubt about the modeling results to be used for the subsequent analysis of lake heatwaves.

REVIEWER COMMENTS

Reviewer #1 (Remarks to the Author):

Heat extremes have been widely awarded by the public, however, its impact on aquatic ecosystems is under studied. In this work, the authors present a quantitative analysis about heat wave impacts on long-term lake warming. While the finding that heat waves contribute to long-term lake warming in China is certainly important for the limnology community, there are several critical aspects of the analysis remains to be justified and improved. Therefore, some more analysis and major revision are necessary.

Response: We thank the reviewer for these positive comments and also for the helpful suggestions that we have used to improve the readability and impact of the manuscript. We have addressed all of the points raised, including the use of a moving baseline, the comparison of results after removing 2022, the reasons for using Landsat data, the lack of clarity in terminology, and quantification.

According to the reviewers' suggestions, we have made the following major changes:

(1) We have changed the title of this manuscript to “The impact of extreme heat on lake warming in China” to avoid any confusion with the previous research paper, and also providing a clearer statement of the research topic.

(2) We have clarified some processes and definitions, such as the calculation of RHLSWT (lake surface water temperature after removing heat extremes) and RHSAT ((lake surface air temperature after removing heat extremes)), to prevent our work from being misleading and confusing.

(3) We have modified the fixed baseline of 1985-2022 for calculating heat extremes in our methods to a partially moving baseline, to exclude the effect of long-term mean-state air temperature and lake surface water temperature (LSWT) change on variation of heat extremes, and have updated the data throughout the manuscript.

(4) We repeated the entire calculation for 1985-2021. Comparing the calculations for 1985-2022, there is little difference (approximately 4%) in the contribution of heat extremes to LSWT, which is useful to consolidate our conclusions.

(5) We have added the sensitivity of the heat extremes of LSWT to changes in the heat extremes of air temperatures.

(6) We have explained the reasons for choosing Landsat data rather than combining Landsat and MODIS data for LSWT simulation.

We hope that the revisions have addressed your most important concerns. Your comments are invaluable in helping us improve our work. Please do not hesitate to ask any follow-up questions, comments or suggestions.

1. I notice that the title “Extreme heat drives unprecedented warming of lakes in China” is very similar to the title of the author’s last paper “A record-breaking extreme heat event caused unprecedented warming of lakes in China” which has been cited by this manuscript. This should be absolutely avoided in scientific publication. I think the first thing is to justify the unique value of this work in comparison to the author’s last paper, and why do not put the two works into single manuscript? The title should be modified accordingly.

Response: Thank you for pointing out the potentially misleading similarity between the title of our manuscript and that of our previous paper. Our previous paper “A record-breaking extreme heat event caused unprecedented warming of lakes in China” analyzed the pattern of change in air temperature and LSWT of 119 lakes larger than 100 km² in China using the NCEI (National Centers for Environmental Information) meteorological dataset and MODIS (Moderate Resolution Imaging Spectroradiometer) land surface temperature data from 2000-2022. It focuses on quantifying the impact of the extreme heat event in June-August 2022 experienced by China, the longest duration and highest intensity since 1961, on surface water temperatures in China’s lakes.

In this study, we used a hybrid physical/statistical-based model to simulate the changes in surface water temperature of China's lakes in the presence and absence of heat extremes. Thus evaluating the effect of heat extremes on surface water temperature in 2260 lakes in China from 1985 to 2022. The study highlights the dominant impact of heat extremes on long-term changes in LSWT, accounting for 36.5% of the warming trend of the studied lakes. These points have been described in the abstract.

Following the reviewer's suggestion, we carefully considered the difference between the two research papers and changed the title of this manuscript to “The impact of extreme heat on lake warming in China”. Thus avoiding any confusion with the previous research paper, and also providing a clearer statement of the research topic.

2. Line 71-84: please specify in front of this paragraph that you are talking about the lake heat waves not heat waves of the atmosphere.

Response: We apologize for the misleading information in the manuscript. In the paragraph at the beginning of Line 71, we want to illustrate the spatial and temporal pattern of the heat extremes of air temperatures in the whole of China for the period 1985-2022. We have therefore added an emphasis on "air temperature" in the first sentence and strengthened the distinction between related concepts throughout the manuscript. Changes have been made in **Line 72, Page 4**.

3. Line 85-86: I do not see the connections between this sentence and the context. Is there any relationship between heat waves in 2022 and the trend of heat wave over the last 38 years? Do you want to emphasize heat wave in 2022 is the only heat wave that exhibited inconsistent spatial pattern with the long-term trend of heat waves? Nevertheless, if you think it is necessary to emphasize the heat wave in 2022, please provide a figure or cite your heat wave paper about its spatial pattern, otherwise it is difficult for readers to follow your argument.

Response: Thank you for pointing out the inconsistency of this sentence in the manuscript. We would like to highlight the magnitude of the intensity and duration of the extreme heat events for air temperatures in 2022, which are prominent throughout 1985-2022. Following the reviewer's suggestion, the paragraph from Line 85 has been merged with the previous paragraph after careful consideration. The description of extreme heat event in 2022 (including comparison of the number of extreme heat days, cumulative heat, and heat intensity in 2022 with those in 1985-2021) has been placed in Result Section for emphasis and prominence. Changes have been made in **Line 147-161, Page 8**.

4. Line 104: “)” is missing.

Response: We apologize for the fault in the writing of this manuscript and thank you for pointing it out. We have added the missing ")") and double-checked the writing and formatting of the entire manuscript. Changes have been made in **Line 103, Page 6**.

5. Line 119-120: Do you mean “after removing heat extremes from SAT in your forcing field of Air2Water model”?

Response: Thank you for bringing this to our attention. We didn't express our point clearly enough. The LSWT was simulated using the surface air temperature (SAT) as forcing field in the Air2Water model, and the best parameters of the model were retained. The LSWT after removing heat extremes (RHLSWT) was calculated using the previously calculated optimal parameters combined with the RHSAT (SAT after removing heat extremes) as the forcing field in Air2Water. So, it's indeed better to be described as "after removing heat extremes from surface air temperature (SAT) in the forcing field of Air2Water" here. Changes have been made in **Line 113, Page 6**.

6. Line 125: I was confused by RHLSWT. Does it mean removing the heat waves from LSWT or removing heat waves from the SAT in your forcing field of Air2Water model? Removing heat waves from the SAT in the forcing field of Air2Water model is an effective way to count contribution of heat wave to long-term trend of lake warming.

Response: Yes, the reviewer is correct. In this manuscript, the LSWT after removing heat extremes (RHLSWT) was calculated in Air2Water using the RHSAT (SAT after removing heat

extremes) as forcing data. We strongly agree with the reviewer that "Removing heat extremes from the SAT in the forcing field of Air2Water model is an effective way to count contribution of heat extremes to long-term trend of lake warming". We have revised misleading statements like this one throughout the manuscript in response to the reviewer's suggestions. Changes have been made in **Line 125, Page 7**.

7. Methods: How is the heat wave calculated? Did the authors use a temporarily constant threshold or a running-smoothed threshold for the heat waves? This work is to count the contribution of heat waves to long-term warming, and threshold of heatwaves should increase along with the increase of background temperature (i.e., mean temperature, because temperature generally follows a normal distribution). Therefore, using a running smoothed threshold to avoid mixture of long-term warming trend and heat wave in the calculation is necessary for this work.

Response: We thank the reviewer for this suggestion. In our manuscript, each day in the extended summer season (June - September) that exceeds the 90th percentile climatology of the corresponding calendar day is defined as a heat extreme day. The 90th percentile climatology was produced by computing the daily 90th percentile of air temperature using a 31-day moving window over a fixed baseline of 1985–2022.

Indeed, by adopting a fixed baseline for the heat extreme analysis, existing literature suggests that heat extremes have become stronger and longer during the past four decades, and many parts of the world are projected to reach a permanent extreme state in the future (i.e., a full year of heat extreme days)^{1,2}. To exclude the effect of long-term mean-state air temperature

and LSWT change on variation of heat extremes, we recalculated the heat extremes using a partially moving baseline according to refs. 3 and 4.

The modified method about the calculation of heat extremes is as follows. Each day in the extended summer season (June - September) that exceeds the 90th percentile climatology of the corresponding calendar day is defined as a heat extreme day. The 90th percentile climatology was produced by computing the daily 90th percentile of air temperature or LSWT using a 11-day window centered on the day of year over a partially moving baseline. The partially moving baseline is fixed for the first and last 31 years of 1985-2022, and moves in the middle. Specifically, the heat extremes for 1985-2000 were calculated using a fixed baseline from 1985 to 2015, for 2007-2022 a fixed baseline from 1992 to 2022, while for 2001-2006, a 31-year moving baseline centered on the year in question was used. Changes have been made in **Line 282-289, Page 14.**

References:

1. Oliver ECJ, Donat MG, Burrows MT, et al. Longer and more frequent marine heatwaves over the past century. *Nat Commun* **9**, 1324 (2018).
2. Pilo GS, Holbrook NJ, Kiss AE, et al. Sensitivity of Marine Heatwave Metrics to Ocean Model Resolution. *Geophysical Research Letters* **46**, 14604-14612 (2019).
3. Hobday AJ, Alexander LV, Perkins SE, et al. A hierarchical approach to defining marine heatwaves. *Progress in Oceanography* **141**, 227-238 (2016).
4. Wang SP, Jing Z, Sun D, et al. A New Model for Isolating the Marine Heatwave Changes under Warming Scenarios. *Journal of Atmospheric and Oceanic Technology* **39**, 1353-1366 (2022).

8. Methods: Line 290-292, as follows the comments above, I would suggest that obtaining RHSAT by replacing heat extreme days with running smoothed temperatures instead of climatological mean of calendar days.

Response: On the basis of the methodology modified in the above response, we utilize the climatological mean calculated from the partially moving baseline as a replacement for the days of heat extremes in the SAT to obtain the RHSAT, instead of using the climatological mean derived from the fixed baseline of 1985-2022.

After modifying the method of calculating heat extremes, the number of extreme heat days experienced by the entire China in 1985-2022 varied from an average of 412 days to an average of 454 days. The trend of number of extreme heat days ranged from 2.62 day/decade to 2.08 day/decade. The highest trend in heat intensity across China ranged from 0.35 °C/decade to 0.29 °C/decade. The average trend in RHSAT ranged from 0.21 °C/decade to 0.22 °C/decade and that in RHLSWT ranged from 0.12 °C/decade to 0.13 °C/decade.

We have updated the calculations of data and figures throughout the manuscript using the new methodology. See **Line 282-289 (Page 14)** for modifications to the methodology.

9. I see the linear trend analysis of heat waves count in 2022, however, heat waves in 2022 are two times stronger than heat waves in 1985-2021 based on Figure 1 a3 and b3. It is hard to say such strong heat wave was totally attributed to long-term trend, instead, inter-annual variability plays a big role in the heat waves of 2022. If excluding 2022 from the heat wave analysis, the

result could be quite different. It is suggested repeat the whole analysis without 2022 and check whether conclusion changed. I believe this will solidify the conclusion.

Response: Thank you for pointing out the abnormalities in the metrics for heat extremes in 2022. We scrutinized all the raw data, computational processes and processing details, and finally realized that it was our acquisition of air temperature in 2022 from ERA5-Land that had problems with spatial and temporal resolution. Referring to the two previous suggestions, we calculated heat extremes for the rectified data using a moving threshold and updated Figure 1 and all data in the main text of the manuscript. It is worth to note that in calculating the heat extremes for 1985-2021, the moving thresholds we used were calculated only up to 2021, which means that the heat extremes in 2006-2021 were all calculated using the 31 years from 1991 to 2021 as the base period (Figure R1). Whereas, when calculating the heat extremes for 1985-2022, the heat extremes for 2007-2022 were calculated using the 31-year base period from 1992 to 2022 (Figure 1).

After correcting the data and updating the method of calculation, the results showed that the average number of days of heat extremes for national air temperatures in 2022 was approximately 26, which is about twice than that in the period of 1985-2021 (approximately 12 days). The average cumulative heat of air temperature in China for 2022 amounted to about 30°C, about three times as much as the 11°C of 1985-2021. The average heat intensity in 2022 is about 25% greater than that of the past 37 years. Therefore, the heat extremes in 2022 were much stronger than the average state in 1985-2021, and repeating the whole calculation for 1985-2021 was necessary for this research.

The trends in the heat extremes in air temperature and LSWT did not differ much between the cases including and excluding 2022 because we used the Theil-Sen method for the calculation, which largely mitigates the impact of extremes similar to 2022 on the long-term trends (Figure 1 and Figure R1). The trend of number of heat extreme days, cumulative heat and heat intensity of air temperature was 2.05 day/decade, 1.88 °C/decade and 0.04 °C/day·decade for 1985-2021, respectively. While the trend in the number of heat extreme days, cumulative heat and heat intensity of air temperature was 2.08 day/decade, 1.95 °C/decade and 0.04 °C/day·decade for 1985-2022, respectively. The average trend of LSWT was 0.16 °C/decade for 1985-2021 and 0.15 °C/decade for 1985-2022 (Figure R2). The average contribution of heat extremes to LSWT was 40.4% for 1985-2021 and 36.5% for 1985-2022. Therefore, the heat extreme in 2022 did not have a significant impact on our results, and these analytical methods described above are robust. Modifications have been added to **Line 147-161, Page 8.**

Figure 1| Spatial and temporal variations in air temperature extremes. a1, a2, and a3 show the total number of heat extreme days from 1985 to 2022, the decadal trend, and curve of annual variation in the five lake regions (Eastern Plain Lake Region, North-east Plain Lake Region, Inner Mongolia-Xin Jiang Lake Region, Tibetan Plateau Lake Region, and Yunnan-Guizhou Plateau Lake Region), respectively. **b1, b2, and b3** indicate the cumulative heat for 38 years, the decadal trend, and the curve of annual variation in the five lake regions, respectively. **c1, c2, and c3** represent the corresponding information of heat intensity.

Figure R1| Spatial and temporal variations in air temperature extremes. a1, a2, and a3 show the total number of heat extreme days from 1985 to 2021, the decadal trend, and curve of annual variation in the five lake regions (Eastern Plain Lake Region, North-east Plain Lake Region, Inner Mongolia-Xin Jiang Lake Region, Tibetan Plateau Lake Region, and Yunnan-Guizhou Plateau Lake Region), respectively. **b1, b2, and b3** indicate the cumulative heat for 37 years, the decadal trend, and the curve of annual variation in the five lake regions, respectively. **c1, c2, and c3** represent the corresponding information of heat intensity.

Figure R2| Comparison of variation in LSWT and the contribution of heat extremes to LSWT for 1985-2022 and 1985-2021. a1, a2, and a3 show the trend of LSWT, RHLSWT and the contribution of heat extremes to LSWT for 1985-2021. b1, b2, and b3 show the trend of LSWT, RHLSWT and the contribution of heat extremes to LSWT for 1985-2022.

10. I would suggest the authors analyze the sensitivity of lake heat wave to the heat wave in the atmosphere, e.g., what is the changing magnitude of lake heat wave intensity corresponding one unit of the accumulative heat of atmospheric heat waves.

Response: Thank you for your thoughtful suggestion. The addition of sensitivity analyses helped to illustrate the response of heat extremes of the LSWT to heat extremes of air temperature, and is a complementary argument to the main topic of this manuscript. We have added a figure to the Supplementary Information analyzing the sensitivity of the heat extremes of LSWT to changes in the heat extremes of air temperature (Supplementary Figure 5).

The results showed that when the annual cumulative heat of air temperature increased by 1 °C, the intensity of the heat extremes of LSWT increased by approximately 0.01 °C/day, the

cumulative heat increases by about 0.45 °C, and the number of days of heat extremes increases by about 0.66 days. When the number of days with heat extremes in air temperature increases by 1 day, the number of days with heat extremes in LSWT increases about 0.93 days, and the cumulative heat gains about 0.51 °C. An increase of 1 °C/day in the intensity of heat extremes in air temperatures would result in an increase in the intensity of heat extremes in LSWT of about 0.42 °C/day. The analysis has been added to Line 117-123, Page 6-7.

Supplementary Figure 5 | Sensitivity of heat extremes of LSWT to heat extremes of air temperature. a-c, Comparison of the three elements of heat extremes (cumulative heat, days

of heat extremes and heat intensity) of LSWT with the cumulative heat change of air temperature. **d-f**, Comparison of the three elements of heat extremes of LSWT with days of heat extremes of air temperature. **g-i**, Comparison of the three elements of heat extremes of LSWT with days of heat intensity of air temperature.

11. I see the authors use Landsat based lake surface temperature data product. Why not use MODIS based lake surface temperature product which has much higher temporal coverage (daily vs monthly from Landsat data) and accuracy, as the authors mentioned the satellite-based observation is very important for the accuracy of data driven model? If the authors worried MODIS data only available in 2001 to present and for large lakes, they can merge the two sets together. The authors may have some important arguments that Landsat-based only products are accurate enough for the analysis, and please justify it in the manuscript.

Response: Thank you for your suggestion. After careful thought and experimentation, we have the following reasons to justify the use of Landsat data as the reference data for Air2Water model.

(1) The spatial resolution of the LST product from Landsat is 120 m, whereas that from MODIS is 1 km. The studied lakes in the manuscript in China have a surface area greater than 1 km². Given the necessity of removing regions of non-permanent water and lake boundary effects (see Methods), there would only be approximately 795 lakes for research if MODIS data were used as the reference for the Air2Water model. In this case, the number of study lakes would be reduced by about two-thirds, considering that we identified 2,260 lakes with Landsat

data in the manuscript, and would thus lacking the investigation of many mid-sized and small lakes.

(2) The reference data for the 795 lakes that meet the requirements for MODIS observations are about seven times larger than those for the remaining 1,465 lakes if MODIS and Landsat data are merged. in the case of Lake Taihu, there are 5,237 reference data after merging, compared to 677 reference data when Landsat is used alone. This great imbalance in the amount of reference data will lead to inconsistencies in the simulation results of Air2Water among these studied lakes.

(3) The LST products from MODIS and Landsat differ significantly in their observations of the same lake on the same day because of the different sensors and algorithms used (Figure R3).

(4) Taking Lake Taihu as an example, we compared 13505 sets of measured LSWT and LSWT simulation results using the merged MODIS and Landsat data (averaged if both sensors provided data on the same day) as reference for Air2Water with those using only Landsat data as reference, respectively. The results showed that the LSWT simulations using Landsat alone as the reference for the Air2Water model were much closer to the measured data.

In summary, we have compared the LST products of MODIS and Landsat in terms of their spatial resolution, temporal resolution, and simulation effect, illustrating the necessity of selecting Landsat alone as the reference for Air2Water model.

Figure R3| Comparison of simulation results using different satellite data as reference. a, Comparison of measured LSWT and LSWT simulated from Air2Water using Landsat as the only reference. **b,** Comparison of measured LSWT and LSWT simulated from Air2Water using merged data from Landsat and MODIS as reference. **c,** Comparison of Landsat and MODIS data on the same day.

12. Reference 7: Instead of citing the research Briefing, I prefer to cite the original research paper, which is entitled “Disruption of ecological networks in lakes by climate change and nutrient fluctuations”.

Response: Thank you for recommending the original research paper as reference to our work. Citing original research paper increases the scientific rigor of our arguments. According to the reviewer's suggestion, we have carefully read the research paper you mentioned, and introduced and cited the paper in the revised manuscript, replacing the original briefing with it. Changes have been made in **Line 336-337, Page 16.**

13. Supplementary Figure 1: I do not see the necessity of illustrating three climatic zones in this figure. There is very little text citing the three climatic zones. Instead, illustrating the extent of the five lake regions is necessary.

Response: Thank you for your suggestion. We have removed the four climatic zones and added five lake regions in the Supplementary Figure 1.

Supplementary Figure 1| Geographic characteristics of the study area. a, Map of studied Lakes. The area of lakes is denoted by the size of circles. The five lake regions in China, namely Eastern Plain Lake Region (I), North-east Plain Lake Region (II), Tibetan Plateau Lake Region (III), Inner Mongolia-Xin Jiang Lake Region (IV) and Yunnan-Guizhou Plateau Lake Region (V) are indicated by different colored borders. The three terraces in China according to elevation are indicated by brown, yellow and green color blocks, respectively. Population density is distinguished by different red blocks. b and c are the counts of lakes along latitude and longitude, respectively

Reviewer #2 (Remarks to the Author):

This study used a data-driven model to assess the contribution of heat extremes to variations in surface water temperature of 2260 lakes in China from 1985 to 2022. The study is of great interests to the broad scientific community. I don't have any other comments except for two concerns about the methods, see below for the details.

Response: We appreciate these positive comments as well as the helpful suggestions from the reviewer, which we have utilized to improve the manuscript. We have offered discussions on your questions about the definition and selection of the model, which we hope can address your questions.

According to the reviewers' suggestions, we have made the following major changes:

(1) We have changed the description of Air2Water model as "a hybrid physically-based/statistical model" to avoid conflicts in the dimension of the model.

(2) We have illustrated the advantages of Air2Water by comparing it with Flake and ANN.

This analysis illustrates that Air2Water is suitable for the current work.

Thank you kindly for your valuable suggestions, which are very beneficial for the improvement of our manuscript.

Methods:

1. "The Air2Water model is a zero-dimensional thermal budget model". I doubt about this as the Air2Water model is not a 0D model. The model is used for LSWT. However, the inclusion of a time-varying δ allows this model to be classified as 0.5D, as it simulates the temporal

evolution of the epilimnion. We described this model in detail in a recent review, which is now under review at Reviews of Geophysics. I suggest the authors rephrasing this sentence.

Response: We appreciate this comment by the reviewer and we have now modified the manuscript accordingly. Specifically, we have revised the manuscript description of Air2Water carefully and scrutinized relevant research papers. Air2Water model was first developed by Piccolroaz et al in 2013¹. In the research paper entitled “Global reconstruction of twentieth century lake surface water temperature reveals different warming trends depending on the climatic zone” in 2020 by Piccolroaz et al², there is a sentence about the Air2Water model: “It is a zero-dimensional heat budget model to the well-mixed surface volume of the lake, accounting for all the heat flux components at the lake-atmosphere interface (shortwave radiation, longwave radiation, and diffusive terms) mathematically simplified to obtain a simple ordinary differential equation”. This is the basis of our previous reference.

However, we agree with the reviewer that the Air2Water model does take into account the temporal evolution of the epilimnion, which is not strictly in line with the criterion that 0-dimensional models do not take into account the variation of the vertical depth. After referring to other research papers by Piccolroaz et al^{3, 4}, as well as other papers related to Air2Water model, we decided to adopt the most commonly used description of Air2Water model, "a hybrid physically-based/statistical model". Changes have been made in **Line 25-26, Page 2 and 202, Page 10**. Thank you for bringing this to our attention.

References:

1. Piccolroaz S, Toffolon M, Majone B. A simple lumped model to convert air temperature into surface water temperature in lakes. *Hydrology and Earth System Sciences* **17**, 3323-3338 (2013).
2. Piccolroaz S., Woolway R. I., Merchant C. J. Global reconstruction of twentieth century lake surface water temperature reveals different warming trends depending on the climatic zone. *Climatic Change* **160**, 427-442 (2020).
3. Piccolroaz S. Prediction of lake surface temperature using the air2water model: guidelines, challenges, and future perspectives. *Advances in Oceanography and Limnology* **7**, 36-50 (2016).
4. Toffolon M, Piccolroaz S, Majone B., et al. Prediction of surface temperature in lakes with different morphology using air temperature. *Limnology and Oceanography* **59**, 2185-2202 (2014).

2. The reason for choosing the air2water model needs justification. I know that the air2water model is simple to use, however, there are some models that have proven to significantly outperform this model, e.g., the stacked machine learning model in “A stacked machine learning model for multistep ahead prediction of lake surface water temperature”. I concern about this point also because of the relatively large MAE values in this study and I doubt about the modeling results to be used for the subsequent analysis of heat waves.

Response: We thank the reviewer for their suggestions on our research methodology. According to the reviewer's suggestion, we have added here a comparison of the Air2Water model with another commonly used lake model (freshwater lake model, Flake) and a widely

used machine learning model (Artificial Neural Network, ANN) in terms of its effectiveness in simulating LSWT.

The FLake model is a one-dimensional bulk model based on the concept of self-similarity, where the vertical profiles of lake ice, the mixed layer, the thermocline, and the thermally active upper layer of sediments are described by their own shape functions¹, which contributes to its low computational cost. It has been widely tested on lakes in China^{2, 3, 4}. The meteorological forcing data used in Flake were obtained from ERA5-Land, and the parameter settings are referenced in Ref. 5.

Yousefi & Toffolon compared a variety of machine learning models for simulating LSWT in 2022 and showed that ANN is the most used and successful ML algorithm for LSWT prediction⁶. The improved machine learning method mentioned by the reviewer can achieve high accuracy predictions of LSWT. However, the amount and discontinuous distribution of our satellite-derived LSWT data are not sufficient for the simulation of LSWT with this method. Therefore, we chose ANN as a representative of the machine learning methods to be compared and analyzed with Air2Water and Flake.

We have compared the simulation results of Flake and ANN for the same air temperature and observed LSWT from Landsat as a reference in six lakes for which *in situ* measured data are available. The results showed that the average R^2 , MAE and RMSE between simulations and *in situ* LSWT for Flake are 0.74, 3.62 °C and 4.61 °C, respectively (Supplementary Figure 9). For ANN, values of the three metrics are 0.87, 2.59 °C and 3.13 °C, respectively. Both Flake and ANN simulations did not show comparable performance to the *in situ* measured data as

Air2Water (R^2 , MAE and RMSE were 0.96, 1.38 °C and 1.75 °C, respectively), demonstrating the applicability of Air2Water model for present research.

As for the issue that the MAE may be too large for heat wave analysis, there are many existing researches where the MAE of the LSWT simulation results compared with the measured data are similar or even larger than the results in this manuscript, but they can be used for the heat wave analysis as well^{5,7}. We strongly agree with the reviewers that the accuracy of the simulation results of LSWT directly affects the analysis of heat waves in lakes, so we would explore methodological models with higher accuracy in our future research. We have added the figure comparing results from the different models to the Supplementary Information, and have added the relevant comparisons to the Methods, at **Lines 241-263, Page 12-13**.

References:

1. Mironov, D., Heise, E., Kourzeneva, E. et al. Implementation of the lake parameterization scheme Flake into the numerical weather prediction model COSMO. *Boreal Environ Res* **15**, 218-30 (2010).
2. Li, X., Peng, S., Deng, X., et al. Attribution of lake warming in four shallow lakes in the middle and lower Yangtze River basin. *Environ Sci Technol* **53**, 12548-55 (2019).
3. Kirillin, G., Wen, L., Shatwell, T. Seasonal thermal regime and climatic trends in lakes of the Tibetan highlands. *Hydrol Earth Syst Sci* **21**, 1895-909 (2017).
4. Wang, X. W., Shi, K., Zhang, Y. L. et al. Climate change drives rapid warming and increasing heatwaves of lakes. *Sci Bull (Beijing)* **68**, 1574-1584 (2023).
5. Woolway, R. I. et al. Lake heatwaves under climate change. *Nature* **589**, 402-407 (2021).

6. Yousefi, A. & Toffolon, M. Critical factors for the use of machine learning to predict lake surface water temperature. *J. Hydrol.* **606**, 127418 (2022).

7. Oliver, E. C. J., Donat, M. G., Burrows, M. T. et al. Longer and more frequent marine heatwaves over the past century. *Nat Commun* **9**, 1324 (2018).

Supplementary Figure 9| Comparison of the accuracy of Flake and ANN. a1-a6, Comparison of LSWT between simulated data from Flake and *in situ* observations in Lake Erhai, Lake Hulunhu, Lake Namco, Lake Luguhu, Lake Qiandaohu and Lake Taihu. b1-b6 are

comparisons of LSWT between simulated data from ANN and *in situ* observations in above six lakes.

REVIEWERS' COMMENTS

Reviewer #1 (Remarks to the Author):

I appreciate the author's response to my comments. I have several other minor comments based on the revisions.

1. I appreciate the author's response. I see that by applying partially moving base line for the heat extreme calculation instead of a fixed base line, contribution of lake heat waves to long-term lake warming changed from 58% to 36.5%. Though its contribution reduces, its role in lake warming is unignorable. However, the word 'dominant' in the abstract is no-longer appropriate to describe its impacts on long-term lake warming. The author may have better wording choice here.

2. Though a concise abstract is feasible for a wide readership. However, the current version of the abstract does not contain enough essential information for the manuscript. I can only get that lake heat waves contribute 36.5 % to lake warming across China from the abstract. However, other essential results, for example drivers of lake heatwaves, changes in the intensity and duration of lake heat waves, are not presented in the abstract.

Reviewer #2 (Remarks to the Author):

The authors added a new part by comparing air2water with Flake and ANN, which is a great improvement to justify the methods. In my opinion, the paper is acceptable now.

REVIEWER COMMENTS

Reviewer #1 (Remarks to the Author):

I appreciate the author's response to my comments. I have several other minor comments based on the revisions.

Response: We appreciate the reviewer's dedication in improving the manuscript, and we have made the manuscript more rigorous and scientific according to these suggestions. The specific changes are listed below.

1. I appreciate the author's response. I see that by applying partially moving base line for the heat extreme calculation instead of a fixed base line, contribution of lake heat waves to long-term lake warming changed from 58% to 36.5%. Though its contribution reduces, its role in lake warming is unignorable. However, the word 'dominant' in the abstract is no-longer appropriate to describe its impacts on long-term lake warming. The author may have better wording choice here.

Response: Thank you for pointing out the inappropriateness of the word "dominant" in the abstract. It is not an accurate representation of the contribution of 36.5 %. Therefore, we have replaced "dominant" with "considerable" according to your suggestion. Changes have been made in **Line 31, Page 2**.

2. Though a concise abstract is feasible for a wide readership. However, the current version of the abstract does not contain enough essential information for the manuscript. I can only get that lake heat waves contribute 36.5 % to lake warming across China from the abstract.

However, other essential results, for example drivers of lake heatwaves, changes in the intensity and duration of lake heat waves, are not presented in the abstract.

Response: Thank you for your valuable suggestion. To complement the important findings of this study, we have included descriptions of the variation in heat extremes in China, as well as the rate of change in lake surface water temperatures before and after the removal of heat extremes in the “Abstract” section. It is “Our study indicates that heat extremes are increasing at a rate of about 2.08 days/decade and an intensity of about 0.03 °C/ day-decade in China. The warming rate of lake surface water temperature decreases from 0.16 °C/decade to 0.13 °C/decade after removing heat extremes.” Changes have been made in **Line 27-30, Page 2.**

Reviewer #2 (Remarks to the Author):

The authors added a new part by comparing air2water with Flake and ANN, which is a great improvement to justify the methods. In my opinion, the paper is acceptable now.

Response: We thank the reviewer for recognizing this research and for all of his/her comments in previous revision, which we have followed to improve the readability and impact of the manuscript.